# Meaning of Respect for Older People in Family Relationships

**DOI:** 10.3390/geriatrics7030057

**Published:** 2022-05-18

**Authors:** Soheila Shamsikhani, Fazlollah Ahmadi, Anoshirvan Kazemnejad, Mojtaba Vaismoradi

**Affiliations:** 1Nursing Department, Faculty of Medical Sciences, Tarbiat Modares University, Tehran 14155-4838, Iran; s.shamsikhani@modares.ac.ir; 2Biostatistics Department, Faculty of Medical Sciences, Tarbiat Modares University, Tehran 14155-4838, Iran; kazem_an@modares.ac.ir; 3Faculty of Nursing and Health Sciences, Nord University, 8049 Bodø, Norway; mojtaba.vaismoradi@nord.no

**Keywords:** family member, home care, older people, respect, qualitative research

## Abstract

Background: Older people have various physical and mental health needs and often receive help from their family members to perform their daily life activities. This research aimed to explore the meaning of respect for older people in family relationships. Methods: A qualitative study using a content analysis approach was conducted. Semi-structured interviews were performed with sixteen older people and four family members. Results: Three main categories were developed: “respect for personal interests”, “kind and sincere respect” and “respect for autonomy”. Understanding of the meaning of respect for older people was influenced by special expectations from family members in terms of meeting personal needs, consideration of preferences and interests and empowerment and support to help preserve older people’s independence and autonomy. Conclusions: Family members should be informed and educated with regard to their expected roles in family relationships, and should consider respect as an important factor affecting older people’s well-being.

## 1. Introduction

The World Health Organization (WHO) reports that the aging population is progressively increasing across the globe. The population of people aged ≥60 years will reach 2.1 billion by 2050, and the majority will live in low and middle-income countries [1]. 

Aging is a natural process and is accompanied by various challenges. The biological process of aging is associated with the degeneration of organs, which results in different physical and mental health problems, including vision and hearing loss, cardiovascular diseases, physical and mental disabilities and musculoskeletal, neurological, gastrointestinal and endocrine disorders. These problems can reduce older people’s ability to independently perform their daily life activities, leading to their dependence on others, particularly their family members [2]. Healthcare providers involved in home care for older people should assess the degree of their clients’ dependency on family members, given the impact of such dependency on the well-being and quality of life of both older people and their families [3,4,5]. 

Dependence on family members is associated with the development of different needs and expectations among older people, including the need for support and respect. Support for older people has a crucial impact on their health and well-being [6,7,8], and the topic has received special attention in recent years in different health-related disciplines, including geriatrics, psychology, sociology, sociopolitical sciences and social medicine. The sources of support for older people vary, based on the immediate sociocultural context and healthcare approaches to aging. For instance, the main source of emotional support for older people in the United States is informal support from friends [9,10], while in Asian transitional countries, including Iran, family members are the main sources of support for fulfilling their expectations and meeting their physical, mental and emotional needs [11,12,13]. 

In general, older people greatly value living in their own private home, and consider it a source of identity and integrity [14]. Living at home positively influences their health, well-being and autonomy [15,16]. On the other hand, dependence and being a burden on family members and relatives can damage older people’s dignity [17]. 

### 1.1. Older People’s Need for Respect

Respect as a main need of older people is an important concept in geriatric care, human rights conventions and bioethics. The United Nations highlights that all human beings have the right to be treated with dignity and respect [18]. Specifically, there is a relationship between being respected and being depended on among older people [19]. Caring situations leading to being dependent on others can be interpreted as disrespectfulness [20]. Conversely, feelings of security, freedom and being hopeful for the future are the consequences of being respected [21]. 

### 1.2. Background in Iran

The growth of the aging population in Iran has been rapid. Older people will constitute 21.7% of the Iranian population by 2050, as 1 in 3 of them will be older than 60 years [22,23]. Older people are considered sources of love and emotional support in Iranian families. Therefore, older people receive care from their family members in their own home, avoiding transition to long-term healthcare settings, such as nursing homes. It is noted that the current healthcare system in Iran cannot coherently meet the holistic needs of older people [24,25]. Additionally, transition from the personal own home to the nursing home environment has been interpreted as the loss of family support and independence in performing daily life routines, leading to negative psychological experiences [26]. Home care helps with the preservation of older people’s dignity and reduces healthcare costs [27]. Therefore, Iranian families often play the role of caregivers for older parents at home in order to fulfil their sacred duties to them in their time of disability [5,13]. Children grow up and leave their older parents to develop their own independent lives, but they preserve family connections with them. Therefore, Iranian older people live on their own and in their own homes, and receive support from their children and family members. Their children provide various types of support for their older parents, take responsibility for managing their life affairs and spend their leisure time with them. 

The meaning of respect from the perspective of older people in family relationships has remained unexplored. Therefore, this study was conducted to explore the meaning of respect for older people in family relationships within the cultural context of Iran. It can have implications for the exploration of home care by family members in other cultures and contexts. 

## 2. Materials and Methods

### 2.1. Study Design

A qualitative study using a content analysis approach was conducted from September 2019 to July 2020 in an urban area of Iran. This research method aims at developing new knowledge and insight into social phenomena [28] through a systematic approach to analyzing textual data, in order to explore patterns in individuals’ perspectives and experiences [29,30].

### 2.2. Participants and Setting

The study participants were sixteen older people living in their own home. Additionally, four children were recruited due to their significant roles in showing respect, based on older people’s accounts as explored in the process of data collection and analysis. It helped improve the depth of data collection with regard to children’s respect for older people in family relationships. 

The participants were chosen using the purposive sampling method with a maximum variation in terms of age, gender and marital status. The first author recruited the older people from city parks and mosques, where they mostly gathered and spent their leisure time. They were requested to provide the contact information of their children, who were consequently invited to participate in the study.

The older people were in the age group of ≥65 years, lived at own home, had some form of family support and had no cognitive impairment. The selection criteria were a willingness to participate in this study and the ability to communicate and share experiences and perspectives about the study phenomenon.

### 2.3. Data Collection

Individual in-depth semi-structured interviews were held by the first author at times convenient for the participants and in their preferred places, such as their own homes, their workplaces or city parks. Given the restrictions and health protocols applied for the COVID-19 pandemic, half of the interview sessions were conducted online via phone call and WhatsApp.

The interviews were started using a pilot-tested, open-ended question about the older people’s daily life, as follows: “how is your life going?” It was continued with the question of “what does respect mean in family relationships from your perspective?” Given their emphasis on the provision of support and empowerment by family members as the symbols of respect in family relationships, branching and probing questions were asked in order to explore this aspect and enhance the depth of data collection: “what kind of support do your family members provide for you?”, “what kind of expectations do you have of them?”, and “will you please provide an example?”

Furthermore, the older people’s children were interviewed to elaborate on respect for older people in family relationships using the following questions: “how do you show respect for your older parents?”, and “what do you do to make them be respected?”. 

The interviews lasted 30–60 minutes, depending on the participants’ tolerance and willingness, and were audio-recorded with their consent. Data collection was continued up to data saturation after 16 interview sessions with the older people and 4 sessions with their children, ceasing when each participant had been interviewed once, and when further data collection did not lead to new findings [31].

### 2.4. Data Analysis and Rigor

Data were analyzed concurrently with data collection through content analysis with an inductive approach consisting of the phases of preparation, organization and reporting [28]. Accordingly, the interviews were transcribed verbatim and were read multiple times to ensure familiarity with the data. A coding matrix was developed, meaning units were identified and the transcriptions were coded. Constant comparison between the codes based on their similarities and differences led to the development of subcategories, which were used for developing categories [28,29].

Rigor was ensured using credibility, conformability, dependability and transferability [32]. Techniques used for fulfilling these criteria were prolonged engagement with the data, maximum variation in sampling, member checking and the provision of detailed descriptions of the research process and findings. For member checking, coded data were provided to some participants and the congruence between their experiences and the generated codes was ensured. They were requested to confirm that the transcriptions and corresponding codes reflected their perspectives as shared during the interviews, leading to authentic and reliable data.

Data analysis was performed by the research team (S.SH., F.A., A.K. and M.V.), who are experienced in qualitative research and elderly care, which adds to the credibility of the analysis. Moreover, field and reflective notes were written to reduce the impact of the researchers’ personal feelings and experiences on the interpretation of findings [31]. The translation of the findings from Farsi to English was performed to the highest quality possible under the supervision of a bilingual translator (M.V.) to ensure its validity. 

This study was reported based on the consolidated criteria for reporting qualitative research (COREQ) (Supplementary file) [33].

### 2.5. Ethical Considerations

The Ethics Committee of Tarbiat Modares University approved this study (decree code: IR.MODARES.REC.1398.140). Permissions for conducting the study were also obtained from this university. At the beginning of the interviews, the participants were provided with clear explanations about the research aim and method. They were assured of their anonymity and data confidentiality. The informed consent form was signed by those participants who were willing to take part in this research, and entered into the data collection phase.

## 3. Results

### 3.1. Demographic Characteristics of the Participants 

The mean age of the older people and their children was 71.7 and 41.25 years, respectively (Table 1). The older people were mostly female (68.8%) and half of them were widowed (50%). 

### 3.2. Meaning of Respect

During the data analysis, 170 initial codes were generated and were used for the development of three main categories, as follows: “respect for personal interests”, “kind and sincere respect”, and “respect for autonomy”. The categories and their relevant subcategories were described using the participants’ direct quotations (Figure 1). The meaning of respect for older people from the participants’ perspectives was influenced by special expectations from family members in terms of meeting personal needs, consideration of preferences and interests, empowerment and support to preserve older people’s independence and autonomy. The participants provided descriptions and examples that indicated what the meaning of respect was from their perspectives, and when respect was shown and when it was not shown.

#### 3.2.1. Respect for Personal Interests

According to the older people’s perspectives, family members should recognize and respect their wishes and try to fulfill their requests. The older people mentioned that family members could show their respect for their older parents by paying attention to their needs, preferences and desires. This encompassed considering the need for travel and for pursuing desired activities, remembering and celebrating important dates and holding funeral ceremonies. 

##### Attention to the Need for Travel

The older people had more leisure time after retirement and were able to pursue their preferred activities. Given the strong family connection between older people and their children, they preferred their children to provide them with opportunities to go on sightseeing trips and pilgrim tours at religious cities.

“My children should understand me and respect my need [for travel]. They sometimes go on a trip, but they do not inform me and do not take me with them. I should always remind them about it [to take me with them” (Participant 4; 80-year-old widowed woman).

“My mother likes going on pilgrim tours. I try to provide her with the opportunity to go on pilgrim tours and financially support her” (P. 13; daughter of a 78-year-old widowed woman).

##### Attention to Preferences and Desires

The older people were satisfied when their family members provided them with opportunity to continue with their desired activities. This type of attention encompassed the consideration of older people’s dignity and respect for them as worthy individuals in the family, despite age-related physical and mental decline.

“My entertainments are to keep birds and take care of my flowers. My children help me follow my favorite activities. They buy food for my birds, buy gardening tools for me and take care of my birds and flowers when I go on a trip. I am satisfied with my children, because they show respect for my preferences” (P. 9; 79-year-old married man).

“My daughter-in-law knows about my favorite foods. Whenever she cooks one of them for her own family, she tastefully and patiently sends a dish to me along with bread and salad. Her taste and precision show her kindness toward me and how much I am valuable to her” (P. 5; 78-year-old widowed woman).

##### Attention to Traditional and Cultural Occasions

Respect was demonstrated by family members when different occasions, such as birthdays, family ceremonies and festivals were remembered through the sending of a small gift or a congratulation message. In terms of their end-of-life expectations, the older people expected their family members to treat their remains with respect and hold a respectful funeral ceremony based on their religious, familial and social norms. 

“My father has an introverted and proud personality and rarely expresses his feelings, but his children should remember him on important occasions. He gets upset if I do not congratulate him on Father’s Day. In occasions and festivals, he wants me to visit him and congratulate him and take flowers and confections” (P. 15; son of a 82-year-old married man).

“On Mother’s Day, I like my children to visit me and bring flowers. Sometimes they bring flowers. I have no need for their financial support, but I expect them to visit me, call me, give me gifts, and show their kindness to me and appreciate me” (P. 6; 72-year-old married woman).

“I do not have many expectations of my family members, but they should not have conflicts after my death over my inheritance. I do not have much money, but I expect them to hold a respectful ceremony for me. Some children substitute a respectful funeral ceremony with supporting charities, but they do not really support charities. I would not like my children to treat me like this” (P. 16; 68-year-old widowed woman).

#### 3.2.2. Kind and Sincere Respect

The older people wanted family members to recognize the value of older people’s lifelong endeavors and verbally and non-verbally appreciate them. When they felt physical and mental exhaustion, family members should treat them patiently, understand their concerns, allocate time to them, listen to their words, not blame them for their mistakes, and sincerely respect their knowledge and experience. Instances of kind and sincere respect included appreciation for their lifelong endeavors and support, showing patience and tolerance and understanding and noticing the older people.

##### Appreciation

The older people’s children had received full support before being separated from the family. The older people ignored their own needs, wishes and interests in order to fulfill their children’s needs and provide them with life and educational opportunities. They had become old now and expected their children to appreciate their lifelong endeavors and support.

“I have done my best to rear my children, but my children do not understand how difficult it has been in the past to manage life. I just want them to understand and appreciate it” (P. 8; 76-year-old widower).

“If I want to splash out, go on trips and go on pilgrim tours, I will soon run out of money. To compensate for the sincere respect paid by my son, I avoid going on trips in order to financially help him have a more comfortable life” (P. 3; 65-year-old widowed woman).

##### Patience and Tolerance

The older people differed from the younger generation in terms of mental status, behaviors and communication styles. Therefore, family members should treat them patiently and show them tolerance. 

“I expect my children to forgive my mistakes, not to insult me and not to admonish me in the presence of others. If I do not respond to the greeting of my daughter-in-law due to missing it, I should not be blamed in her presence” (P. 9; 79-year-old married man).

“Whenever I want to go to the doctor, I prefer to be accompanied by my son because his presence is very impatient. If I give him a wrong address, he nags me. I hate nagging. They [children] have forgotten how patiently their parents have treated them when they have been kids” (P. 6; 72-year-old married woman).

##### Acknowledgment of Values and Needs

The older people liked their family members to show respect for their values, devote time to them every week, eagerly listen to them and pay attention to their age-related concerns, such as loneliness. They mostly wanted their children and grandchildren to communicate with them with interest and eagerly listen to their experiences and memories.

“My father loves speaking to others and to be listened to and noticed respectfully. He shares eagerly his memories of the military service and his employment. He really likes all of us to sit and listen to his words and state that we feel proud of him. He may not notice that some of these memories and experiences are repetitious for us” (P. 12; son of a 73-year-old married man).

“My children are in the adulthood period and they should understand me. I am afraid at night. I expect them to be with me at nights so I feel less lonely. I want their respect. They should communicate with me instead of spending their leisure time with their mobile phones. It seems that they barely tolerate me” (P. 20; 66-year-old widowed woman).

#### 3.2.3. Respect for Autonomy

The older people had the feeling of being a burden on family members and receiving mandatory respect. They preferred to have an independent life and gained a sense of self-worth from having the freedom to make their own life decisions. Family members should respect their opinions, provide them with the requirements for an independent life, empower them to use technology and let them independently manage their own financial affairs.

##### Respect for Opinions

The older people preserved their own dignity by having control over their own life affairs and having adequate power to make decisions independently. Their family members should involve them in making major familial decisions to help maintain their sense of importance and worthiness.

“My father has mental and memory impairment, but I still consult with him whenever I want to make important decisions for my own family and my children such as changing my house. I do so to maintain his morale and sense of importance” (P. 15; son of a 82-year-old married man).

“I have a piece of land and I would like to sell it to have a more comfortable life. But my older son does not allow me to do so with the justification that the land should be kept as a source of money for the funeral ceremony” (P. 17; 71-year-old widowed woman).

“I would like to buy a new house, but whenever I consult with my son, he says that I should wait and not to hurry. I do not know what is in his mind and what he conceals from me” (P. 3; 65-year-old widowed woman).

##### Support for an Independent Life

Fulfilling life responsibilities gave the older people a sense of self-sufficiency and hope. They still preferred to host large family parties. To provide the older people with the opportunity for an independent life, their family members should simplify daily life tasks for them, create a supportive environment and directly and indirectly support them.

“I live alone in my own home and whenever I face a problem, I call my children. I am illiterate. My younger daughter has written important phone numbers in a phonebook with paintings next to them instead of writing names. For example, as my older son wears glasses, she has drawn glasses next to his number for me to notice it” (P. 19; 79-year-old widowed woman).

“One of my girls, controls everything in my home and facilitates the environment to help me perform my daily life tasks. She has even threaded several needles so that I can darn my stocks if they are torn. I like to independently do my own tasks and meet my own needs” (P. 8; 78-year-old widower).

“My mother values home cleanliness by me. But she cannot perform all things as before due to limb pain. Since she still wants to maintain her authority and pride, she claims that she is still able to stand on her own feet” (P. 12; daughter of a 78-year-old married woman).

“I am still on my own feet. In the winter, I go shopping if the ground is not slippery. I need the help of family members just for buying heavy things” (P. 2; 72-year-old married man).

##### Empowerment in the Use of Technology

The older people did not have sufficient self-confidence and self-efficacy for the use of technology, due to the fear of making mistakes. Therefore, family members should attend to their need to learn about technology through respectful, simple and frequent training. This helps them preserve a sense of respect and dignity.

“I have adequate monthly income but it is saved in the debit card, which is held by my son. I should ask him to buy necessary things that I need in his spare time. If I could use my card, I could buy whatever I wanted and whenever I wanted” (P. 14; 67-year-old married woman).

“I do not know how to use an ATM machine for paying my utility bills and hence, I should ask for help from my children or neighbors. My children should come with me to help with paying my bills to overcome my fear” (P. 17; 71-year-old widowed woman).

“My mother repeatedly said that she could not use Telegram^®^ and WhatsApp^®^. We bought a good mobile for her and now she is able to make video calls with my sister who lives in another town and with my aunt who lives in another country. Ever since she has become interested in technology and feels happy with it” (P. 13; daughter of a 78-year-old widowed woman).

##### Independent Management of Financial Affairs

The older people preferred to pay for their own expenses and independently manage their own financial affairs. 

“I have not become very poor. I attempt to visit those relatives of mine who get married or deliver a new baby and give them gifts. It makes me to feel proud of myself among my relatives” (P. 4; 80-year-old widowed woman).

“I am very happy when I do not need my children’s financial support and am able to financially support my own affairs. My financial independence preserves my dignity” (P. 2; 72-year-old married man).

## 4. Discussion

This study aimed to explore the meaning of respect for older people in family relationships. Inclusion of the perspectives of the older people’s children helped with the interpretation of their older parents’ perspectives, and in combination, they provided a more complete picture of the meaning of respect. The research findings were presented using three categories: “respect for personal interests”, “kind and sincere respect”, and “respect for autonomy.”

The meaning of respect for older people in family relationships was mostly consisted of the provision of support, and empowerment offered by family members. Support from family members preserved the older people’s dignity through preservation of the feeling of being a worthwhile person in the family, and helped them retain their independence and autonomy to manage their own daily life affairs. Our findings are in line with the notion of respect for human beings in bioethics, with the distinct core concerns of autonomy, dignity, integrity and privacy [34].

Regarding respect for personal interests, the older people expected their family members to pay attention to their need for travel and leisure time. Generally, older people are interested in traveling, prefer to go on long trips and are more worried about their safety than younger people [35]. Their perceived ability to go on trips significantly affects their life satisfaction, as well as their sensory and cognitive experiences [36]. Therefore, family members should pay careful attention to the appropriateness and meaningfulness of trip experiences in order to support the dignity of their older relatives [37]. An active lifestyle and involvement in social activities provides them with a feeling of being alive and healthy [38,39]. Older people with different levels of physical functionality have various preferences in maintaining an active lifestyle, including walking, gardening, doing sports, watching TV, attending religious ceremonies and traveling [40,41]. Their preferences should be considered when strategies are designed to satisfy their need for respect and to reduce their feeling of loneliness [42]. 

Our study findings showed that the older people defined respect in family relationships as attention to their preferences and participation in familial occasions. It is believed that appropriate planning for life activities based on older people’s abilities and preferences is associated with improved feelings of efficiency, self-esteem, pleasure, comfort, creativity and autonomy [43,44]. Older people in various cultures commonly wish that children would pay greater attention to their parents and give them gifts on special occasions. However, the nature of the desired gifts may vary according to the immediate context [45]; for instance, in Korea, gifts are divided into tangible types, such as clothes or money and intangible types, such as the opportunity to give a lecture [46].

The older people in this study highly valued their children’s attention to their funeral ceremony and considered it a matter of respect. Similarly, a study in Turkey showed that one of the main methods of showing respect for older adults was the traditional funeral ceremony [47]. However, changes in the sociodemographic characteristics of populations, mortality patterns and causes of death due to industrialization, urbanization and economic development have changed funeral rituals and the relationship between death and social structure [48]. For instance, the number of post-death worshipping days in Korea has reduced, and the funeral ceremony has become simpler [46]. Some people in Ghana are mainly concerned with the financial costs of the funeral ceremony and its impact on personal and familial respect [48]. In the United States, the funeral ceremony is said to be helpful for family members in coping with loss, though some people may consider it futile [49]. 

Regarding kind and sincere respect, the older people considered that showing appreciation was represented respect for them. Valuing older adults’ experiences, sayings and comments were mentioned as examples of showing respect for older people. In the international literature, obedience to older adults’ orders and comments, besides listening to their words and sharing perspectives with them, have been emphasized as ways of showing respect [46,49]. Appreciation is a positive feeling expressed by providing others with intentional help and love [50], showing admiration, behaving well and exhibiting gratitude [51]. Feeling appreciated can reduce the feeling of loneliness and improve self-reported well-being [52,53] as well as health-related quality of life [54].

In this study, the participants emphasized showing respect for older people’s opinions and empowering them by family members to perform daily life tasks. They liked to share their valuable experiences and to be consulted in decision making. A study on the perspectives of older adults in eleven European countries found that respect meant facilitating older people’s involvement in selfcare and provision of clear information and support for their autonomy [55]. Since older people are often concerned with age-related dependence resulting from physical and mental problems [56], their family members should appreciate their identity and actively involve them in decision making and thereby protect their dignity [57,58].

Our research findings also showed that respect involved empowering older people to take care of their own financial affairs and to use technology and maintain their autonomy. Encouragement and support in using new technology helped older adults ensure their own safety and promote their self-perceived welfare [59,60]. Financial independence and not being a burden on others have significant implications for older people’s sense of autonomy and dignity [57]. 

This study was conducted in an urban area of Iran, but the research findings demonstrated similarities with those of other studies, which enhances our confidence in their transferability to other cultures and contexts. The strength of this research lies in the participation of older people and their family members, which provided a more comprehensive picture of the study phenomenon. More studies should be carried out to improve our understanding of respect for older people in family relationships. 

## 5. Conclusions

Respect for older people in family relationships meant consideration of their personal interests and desires and empowerment and the provision of support to preserve their dignity, independence and autonomy. Exploration of the experiences and subjective perceptions of older people with regard to respect provides valuable data for family caregivers and healthcare providers working to improve the quality of home care and prevent feelings of loneliness among older people. 

Given the significance of family members’ behaviors in the improvement of the quality of life and well-being of older people, they should always be involved in developing initiatives aimed at the improvement of older people’s well-being in home care. Family members should be informed and educated with regard to their expected roles in family relationships, and should consider respect as an important factor affecting older people’s well-being.

## Figures and Tables

**Figure 1 geriatrics-07-00057-f001:**
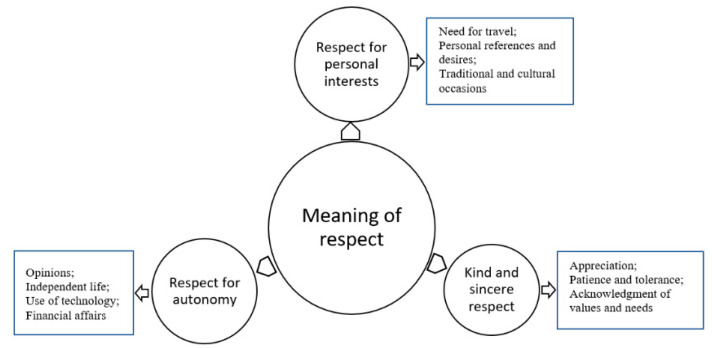
Meaning of respect for older people in family relationships from the participants’ perspectives.

**Table 1 geriatrics-07-00057-t001:** Demographic characteristics of the participants.

Variable	All Participants	Older People	Family Member
Age (y), mean (SD)	65.6 (13.54)	71.7 (5.1)	41.25 (6.23)
Gender, n (%)			
Female	7 (35.0)	11 (68.8)	2 (50.0)
Male	13 (65.0)	5 (31.3)	2 (50.0)
Marital status, n (%)			
Married	9 (45.0)	7 (43.8)	2 (50.0)
Widowed	10 (50.0)	8 (50.0)	2 (50.0)
Widower	1 (5.0)	1 (6.3)	0

## Data Availability

Restrictions apply to the availability of data from this research because of the anonymity of the participants and confidentiality matters.

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
