# Peer review of "Meaning of Respect for Older People in Family Relationships"

_geriatrics, 2022, doi:10.3390/geriatrics7030057_

Round 1

Reviewer 1 Report

Thank you for this interesting article. 

Introduction:

In the first paragraph (line 25) the data is regarding 80+ years old but your study deals with younger people. The data should fit the study population.

Your literature review is missing the contextual characteristics of Iranian society regarding older adults. This lack of context is damaging the understanding and ability of interpretation of study finding. For Example: financial status and shortage of money are well spoken in your citations but there is no mention of it in the literature review. Same goes for living conditions, leisure preferences etc. 

Method

In paragraph 2.2 it is not clear how the adult children were chosen to participate in study. Is it due to something aged interviewees mentioned? if so - What was it?

In addition, it is not well emphasized that they were older interviewees children.

In line 98 - The first question need to be rephrased. 

Line 119 - Congruences between experiences and generated codes was ensured. How?

Paragraph 2.5 - Which University's ethic committee gave the approval?

Results

Overall, context of citations and their content should be added and not only citations. 

Table 1: Authors should consider interviewees functional condition.

Line 163 - Spelling.

Paragraph 3.2.1.1 it is unclear why people after retirement expecting their family members to take them on trips and other leisure activities. Is it a cultural issue? 

Line 217: "I hardly worked"  - I think it should be rephrased to I worked hard in order to get the right citation and context.

Last citation in this paragraph (lines 220-222) is opposite - From parent to child. How does it manifests appreciation? it should be explained.

Paragraph 3.2.2.3 last citation - If children has little kids, why the aged mother expect her children to be with her and not with their kids? It should be explained. 

Paragraph 3.2.3 tackles autonomy: "...They tended to have an independent life and felt worthy at having freedom in making their choices..." There is no explanation regarding aged people autonomy in Iran. Are they autonomous?  In which phase of their lives they lost it ? Are they expected to be autonomous?

Discussion

Line 387 - Abuse is mentioned for the first time. This is a major issue that haven't been discussed during literature review nor findings. If authors want to mention it - It should ne added along article. 

It is not clear whether there have been differences between aged people and older children in the way they comprehend respect.

References

References 1 and 3 are both authored by WHO and yet they are written differently.

Author Response

Dear Editor, Geriatrics

Thank you for the provision of this opportunity to revise the article and resubmit it to your journal. We appreciate the comments of the reviewers by which we improved the quality of the presentation of our article. We considered the comments with care and tried to revise the article accordingly. The changes were highlighted in red color in the text and the details of changes can be found as follows. We hope that the changes meet your journal’s needs. Sincerely Yours/Authors

Comments by Reviewer 1

Introduction:

In the first paragraph (line 25) the data is regarding 80+ years old but your study deals with younger people. The data should fit the study population.

Response: The data was matched for the age group in our research.

Your literature review is missing the contextual characteristics of Iranian society regarding older adults. This lack of context is damaging the understanding and ability of interpretation of study finding. For Example: financial status and shortage of money are well spoken in your citations but there is no mention of it in the literature review. Same goes for living conditions, leisure preferences etc. 

Response: Some more information on the cultural background of older people care in Iran was added, as you wished.

Method

In paragraph 2.2 it is not clear how the adult children were chosen to participate in study. Is it due to something aged interviewees mentioned? if so - What was it?

Response: The recruitment process for adult children was described with more details.

In addition, it is not well emphasized that they were older interviewees children.

Response: It was described with more details that they were family members.

In line 98 - The first question need to be rephrased. 

Response: It was reworded.

Line 119 - Congruences between experiences and generated codes was ensured. How?

Response: As in line with member checking, the brief report of the interviews and related codes were given to the participants and they were asked to give feedback on the transcriptions and that the codes reflected their perspectives during the interviews. It was described in the text.

Paragraph 2.5 - Which University's ethic committee gave the approval?

Response: Tarbiat Modares Univetity.

Results

Overall, context of citations and their content should be added and not only citations. 

Response: The background for citations were described with more details.

Table 1: Authors should consider interviewees functional condition.

Response: The general description of their functional status was added.

Line 163 - Spelling.

Response: It was checked.

Paragraph 3.2.1.1 it is unclear why people after retirement expecting their family members to take them on trips and other leisure activities. Is it a cultural issue? 

Response: Yes, this is a cultural aspect that has been stated in the Introduction.

Line 217: "I hardly worked"  - I think it should be rephrased to I worked hard in order to get the right citation and context.

Response: It was rephrased.

Last citation in this paragraph (lines 220-222) is opposite - From parent to child. How does it manifests appreciation? it should be explained.

Response: This is so, but this citation shows that the older adults try to prevent being burden on their families as much as they can and in response, they expect for their respect.

Paragraph 3.2.2.3 last citation - If children has little kids, why the aged mother expect her children to be with her and not with their kids? It should be explained. 

Response: The older adults do not mean that their children should be always with them. They expect that their children do not leave them alone and spend more time with them to prevent the feeling of loneliness.

Paragraph 3.2.3 tackles autonomy: "...They tended to have an independent life and felt worthy at having freedom in making their choices..." There is no explanation regarding aged people autonomy in Iran. Are they autonomous?  In which phase of their lives they lost it ? Are they expected to be autonomous?

Response: Related information was added to the introduction section with regard to their autonomy.

Discussion

Line 387 - Abuse is mentioned for the first time. This is a major issue that haven't been discussed during literature review nor findings. If authors want to mention it - It should ne added along article. 

Response: It was deleted to prevent misperceptions.

It is not clear whether there have been differences between aged people and older children in the way they comprehend respect.

Response: The meaning of respect from the perspectives of older people and their children were explored in this qualitive research. Inclusion of the both parties’ perspectives would provide the complete description of the meaning of respect. They complete each other. It was stated in the discussion as you wished.  

References

References 1 and 3 are both authored by WHO and yet they are written differently.

Response: The citations were checked.

Sincerely Yours

Authors

Reviewer 2 Report

Review of article titled: Feeling of respect among older adults in family relationships

Overall: The translation to the English language needs improvement throughout the entire manuscript.  I consider this the responsibility of the authors and as a reviewer I will NOT be making these changes. 

Introduction: Just need to mention that the older population is growing – specifics for various countries not necessary – only country that the study is taking place.  As this is a qualitative study and the sample size is small the translation to other cultural groups would need much more study.

Line 72: The authors state that the study is: “qualitative content analysis study”.  Content analysis is a technique that is used to analyze the data of a qualitative study. 

Line 78: The authors state that they obtained a purposeful sample.  More detail on how the participants were recruited would be helpful to help understand who they are. 

Review of the results of the study:

Figure presented line 148: The authors use “older adults’ feelings of respect” as the focus of the study.  Consider making this more specific as noted in the purpose of this study – “older adults’ perspective of the meaning of family member respect” or as noted later in the manuscript “family relationships”.  The authors also use the term “expectations”.  The use of this term does add to the comprehensiveness of the results.  Many of the quotes provide statements of what the older adults expects from the family members.

Line 152: The authors state “Given the long duration of family living under the same roof”.  This is the first time the authors state that the participant and the family lived in the same home – and as I read further on I am not sure of the accuracy of this statement by the author.  Sounds like some do not live with their family.

General comment regarding the reporting of the results using the quotes provided: Quotes that are included many times are describing what is “not” respect by the family.  Authors need to explain to the reader that the older adult provided examples that both described when respect was shown and when it was not shown.

Comments on overall discussion:

The focus of the study is on “respect”.  Yet in the quotes as well as identified by the authors in the discussion other terms are more represented such as support and empowerment.  These terms are important but are not respect.  It is unclear if the interview process included the provision of the definition of respect to help the older person concentrate on that meaning.  The authors need to be deliberate in describing the how the concepts are connected.

In the introduction the authors provide the following statement in line 59 “Respect consists of five main components of autonomy, dignity, integrity, privacy, and vulnerability”.  This statement should be an integral part of the results and the discussion.  How are these 5 components reflected in the statements of the participants?  If not represented, why were they not present? 

Comment on citations:

Unsure if the citations are used accurately in the statements by the authors.   I will provide one example but did note other occurrences of this concern. The following sentence is supported by reference 57.  “Specially, financial independence and not being a burden to others have significant implications for their autonomy, dignity and prevention of abuse in caring relationships [57]”.  This reference is focused on older people with memory disorders – this is not reflected in the text that uses this citation.

Välimäki, T.; Mäki-Petäjä-Leinonen, A.; Vaismoradi, M. Abuse in the caregiving relationship between older people with 542 memory disorders and family caregivers: A systematic review. J Adv Nurs 2020, doi:10.1111/jan.14397.

Author Response

Dear Editor, Geriatrics

Thank you for the provision of this opportunity to revise the article and resubmit it to your journal. We appreciate the comments of the reviewers by which we improved the quality of the presentation of our article. We considered the comments with care and tried to revise the article accordingly. The changes were highlighted in red color in the text and the details of changes can be found as follows. We hope that the changes meet your journal’s needs. Sincerely Yours/Authors

Comments by Reviewer 2

 Overall: The translation to the English language needs improvement throughout the entire manuscript.  I consider this the responsibility of the authors and as a reviewer I will NOT be making these changes. 

Response: We rechecked the text and for sure this is our responsibility to provide understandable text. The cultural-contextual identity of our findings requires that the authors take care of the presentation quality.

Introduction: Just need to mention that the older population is growing – specifics for various countries not necessary – only country that the study is taking place.  As this is a qualitative study and the sample size is small the translation to other cultural groups would need much more study.

Response: We improved this section to indicate that aging is an international trend and our country has similar aging trend as others.

Line 72: The authors state that the study is: “qualitative content analysis study”.  Content analysis is a technique that is used to analyze the data of a qualitative study. 

Response: This is the traditional style of the presentation of a qualitative research method. By the way, we rephrased it as you wished.

Line 78: The authors state that they obtained a purposeful sample.  More detail on how the participants were recruited would be helpful to help understand who they are. 

Response: It was described with more detail.

Review of the results of the study:

Figure presented line 148: The authors use “older adults’ feelings of respect” as the focus of the study.  Consider making this more specific as noted in the purpose of this study – “older adults’ perspective of the meaning of family member respect” or as noted later in the manuscript “family relationships”.  The authors also use the term “expectations”.  The use of this term does add to the comprehensiveness of the results.  Many of the quotes provide statements of what the older adults expects from the family members.

Response: We made the text more clear and preferred to consistently use ‘’older adults’ perspective of the meaning of respect in family relationships.’’

Line 152: The authors state “Given the long duration of family living under the same roof”.  This is the first time the authors state that the participant and the family lived in the same home – and as I read further on I am not sure of the accuracy of this statement by the author.  Sounds like some do not live with their family.

Response: It refers to the time that the whole family lived together under the same roof. But at the time of data collection, our participants were living in their own homes.

General comment regarding the reporting of the results using the quotes provided: Quotes that are included many times are describing what is “not” respect by the family.  Authors need to explain to the reader that the older adult provided examples that both described when respect was shown and when it was not shown.

Response: It was mentioned at the beginning of the results that our quotations covered both of them.

Comments on overall discussion:

The focus of the study is on “respect”.  Yet in the quotes as well as identified by the authors in the discussion other terms are more represented such as support and empowerment.  These terms are important but are not respect.  It is unclear if the interview process included the provision of the definition of respect to help the older person concentrate on that meaning.  The authors need to be deliberate in describing the how the concepts are connected.

Response: This study provided the opportunity to the older adults to define respect as it was in line with inductive approach to our research. What the older adults presented here meant respect from their perspectives, even if they may mean empowerment or support from our (readers) perspectives. It was pointed to at the discussion section.

In the introduction the authors provide the following statement in line 59 “Respect consists of five main components of autonomy, dignity, integrity, privacy, and vulnerability”.  This statement should be an integral part of the results and the discussion.  How are these 5 components reflected in the statements of the participants?  If not represented, why were they not present? 

Response: This is the result of another study. It was transferred to the discussion section.

Comment on citations:

Unsure if the citations are used accurately in the statements by the authors.   I will provide one example but did note other occurrences of this concern. The following sentence is supported by reference 57.  “Specially, financial independence and not being a burden to others have significant implications for their autonomy, dignity and prevention of abuse in caring relationships [57]”.  This reference is focused on older people with memory disorders – this is not reflected in the text that uses this citation.

Välimäki, T.; Mäki-Petäjä-Leinonen, A.; Vaismoradi, M. Abuse in the caregiving relationship between older people with 542 memory disorders and family caregivers: A systematic review. J Adv Nurs 2020, doi:10.1111/jan.14397.

Response: The citations all have been used correctly in the text. This specific citation was used in reference to the significance of independence and not being burden to families that can be generalized to all older people with various mental and cognitive levels. It puts our research findings in a broader perspective in which feeling of respect by the participants is opposed to being abused. Having no financial independence can mean also as abuse. To prevent misperception, this specific citation was replaced by another one.

Sincerely Yours

Round 2

Reviewer 2 Report

I appreciate the changes the authors have made in the newest version of the manuscript.  I have attached a copy of the manuscript with my edits and comments. 

Major comment:  The authors need to make changes in the results section - at the beginning of each meaning section.  I made comments in several of these sections on the attached copy but as it was becoming repetitive I only made comments in a couple of the sections.  The wording currently reflects "what the family should do" - and in the results sections, it should reflect what the older adult said.

Author Response

Dear Reviewer,

Thanks for your comments. The suggestions were incorporated into the text.

Sincerely Yours,

Authors